# Structural Elucidation of a Glucan from *Trichaster palmiferus* by Its Degraded Products and Preparation of Its Sulfated Derivative as an Anticoagulant

**DOI:** 10.3390/md21030148

**Published:** 2023-02-24

**Authors:** Haiqiong Ma, Qingxia Yuan, Hao Tang, Hongjie Tan, Tingting Li, Shiying Wei, Jinwen Huang, Yue Yao, Yaping Hu, Shengping Zhong, Yonghong Liu, Chenghai Gao, Longyan Zhao

**Affiliations:** 1Institute of Marine Drugs, Guangxi University of Chinese Medicine, Nanning 530200, China; 2Guangxi Key Laboratory of Marine Drugs, Guangxi University of Chinese Medicine, Nanning 530200, China

**Keywords:** brittle star, polysaccharide, glucan, structure, anticoagulant activity

## Abstract

Echinoderms have been attracting increasing attention for their polysaccharides, with unique chemical structure and enormous potential for preparing drugs to treat diseases. In this study, a glucan (TPG) was obtained from the brittle star *Trichaster palmiferus*. Its structure was elucidated by physicochemical analysis and by analyzing its low-molecular-weight products as degraded by mild acid hydrolysis. The TPG sulfate (TPGS) was prepared, and its anticoagulant activity was investigated for potential development of anticoagulants. Results showed that TPG consisted of a consecutive α1,4–linked D-glucopyranose (D-Glc*p*) backbone together with a α1,4–linked D-Glc*p* disaccharide side chain linked through C-1 to C-6 of the main chain. The TPGS was successfully prepared with a degree of sulfation of 1.57. Anticoagulant activity results showed that TPGS significantly prolonged activated partial thromboplastin time, thrombin time, and prothrombin time. Furthermore, TPGS obviously inhibited intrinsic tenase, with an EC_50_ value of 77.15 ng/mL, which was comparable with that of low-molecular-weight heparin (LMWH) (69.82 ng/mL). TPGS showed no AT-dependent anti-FIIa and anti-FXa activities. These results suggest that the sulfate group and sulfated disaccharide side chains play a crucial role in the anticoagulant activity of TPGS. These findings may provide some information for the development and utilization of brittle star resources.

## 1. Introduction

Marine organisms such as algae, mollusks, and echinoderms contain various compounds, including carbohydrates, lipids, proteins, and other bioactive molecules, and have great potential for medical applications [1]. Notably, numerous marine polysaccharides present various structures and have potent biological activities such as anticoagulation, anti-inflammation, and anti-tumor [2,3]. For example, hyaluronic acid, chondroitin sulfate, alginate, and chitin are found in marine organisms and have been widely applied in pharmaceutical, biotechnological, and medical fields [4,5]. 

The phylum Echinodermata includes more than 7000 species of widely distributed sea cucumbers, sea stars, crinoids, sea urchins, and brittle stars [6]. Some polysaccharides, such as fucosylated glycosaminoglycans and sulfated fucans, with unique structures, have been isolated from sea cucumbers and have attracted increasing attention due to their excellent anticoagulant and antithrombotic activities [7,8]. Their derivatives prepared by structural modification may serve as a novel promising anticoagulant, without adverse effects such as serious bleeding consequences [8,9]. Brittle stars are widely distributed in deep- and shallow-water marine habitats from the Antarctic and Arctic to the tropics [10]. However, studies on polysaccharides from this species are very limited compared with those of other echinoderms [8,11]. Although there are some studies on saponins, terpenoids, and steroids from brittle stars, only one report on the chondroitin sulfates/dermatan sulfates can be identified [10]. A basic understanding of polysaccharides from brittle stars is essential for their application in drugs, functional foods, and even biomaterials.

Glucans, a homopolysaccharide, have received extensive attention due to their various structures and biological activities, including excellent immunomodulatory and antitumor activities [12,13]. According to previous studies, the biological activities of glucans highly depend on their physicochemical properties and structural features [13]. Structural modifications such as sulfation, phosphorylation, and acetylation can change their physicochemical properties, such as molecular size and water solubility, enhance their pharmacological activities, and even generate new biological activities [14]. For example, sulfated modification increased the anticoagulant activity of a β-(1→6)-D-glucan [15]. 

In the present study, a new glucan (TPG) was obtained from the brittle star *Trichaster palmiferus*, and its structure was elucidated by analyzing its depolymerized products. The TPG has a consecutive α1,4–linked D-glucopyranose (D-Glc*p*) backbone and an α1,4–linked D-Glc*p* disaccharide side chain linked through C-1 to C-6 of the backbone. We hypothesized that the sulfated TPG (TPGS) with a sulfated disaccharide side chain may have potent anticoagulant activity and different mechanisms compared with the low molecular weight heparin (LMWH). Therefore, the TPGS was prepared, and its anticoagulant activity and action mechanism were investigated. This study may provide significant information for developing brittle star glucan as a potential anticoagulant drug.

## 2. Results and Discussion

### 2.1. Extraction, Purification, and Physicochemical Properties of TPG 

Crude polysaccharides were obtained from *T. palmiferus* after enzymatic digestion and alkaline treatment, with a yield of 0.42% by dry weight of the brittle stars. TPG was obtained with a yield of 4.2% by dry weight of crude polysaccharides after purification by the Amberlite FPA98 Cl ion-exchange resin eluted with distilled water, and by the Sepharose CL-6B column (Appendix A). As shown in Figure 1A, TPG yielded a symmetrical and single elution peak on the Shodex OH-pak SB-804 HQ column, confirming that it was a homogeneous polysaccharide. After calibration with standard D-series dextrans, the weight-average molecular mass (*M*_w_) of TPG was determined to be 165.5 kDa. The carbohydrate content of TPG was calculated as 92% by the phenol-sulfuric acid method. There were no absorptions at 260–280 nm in the UV spectrum (Appendix A), indicating the absence of proteins, nucleic acids, and peptides in TPG after isolation and purification [16]. The monosaccharide compositions of TPG are shown in Figure 1B. TPG was a glucan, as only D-glucopyranose (D-Glc*p*) existed in this polysaccharide [17].

The Fourier transform infrared (FT-IR) spectrum of TPG is presented in Figure 1C and assigned according to our previous study [12]. There was a broad and strong absorption peak at 3368.50 cm^−1^, which was attributed to the stretching vibration of O-H. The weak absorption peak at 2935.44 cm^−1^ was caused by the stretching vibration of C-H. The absorption of 1647.22 cm^−1^ was ascribed to the absorption peak of crystal water. The bands at 1154.42 cm^−1^, 1079.90 cm^−1^, and 1023.36 cm^−1^ were due to the pyranose form of sugars. The signal at 931.80 cm^−1^ was assigned to α-glucopyranose derivative. Therefore, TPG may be an α-type glucan consisting of pyranose sugar residues.

### 2.2. Elucidation of the Precise Structure of TPG

The connection patterns of sugar residues in the TPG were determined by methylation analysis according to the literature [18] and are summarized in Table 1. The partially methylated alditol acetate (PMAA) derivatives of TPG were determined to be 2,3,4,6-Me_4_-Glc*p*, 2,3,6-Me_3_-Glc*p*, 2,6-Me_2_-Glc*p*, and 2,3-Me_2_-Glc*p*, based on their retention times and mass fragments. Their peak area ratios were 17.2%, 69.66%, 0.61%, and 12.53%, respectively. These results suggested that TPG contained (1→), (1→4), (1→3,4), and (1→4,6)-linked-Glc*p*, and may have (1→4)-linked-Glc*p* backbone branched at C-6, which was further confirmed by the NMR spectra.

The high molecular weight of TPG resulted in many overlapping and broad signals in its 1D/2D NMR spectra (Figure 2), which seriously hampered analysis of the detailed structures. It is well known that some glycosidic bonds in polysaccharides are easily hydrolyzed under acidic conditions. Therefore, the mild acid hydrolysis is a commonly used method for preparing depolymerized products to investigate structures of the native polysaccharides [19]. In this study, the depolymerized product dTPG was obtained by mild acid hydrolysis using trifluoroacetic acid (TFA). Three fractions (F1–F3) with *M*_w_ of 17.1 kDa, 9.2 kDa, and 1.7 kDa were isolated by a Bio-Gel P-6 column with yields of 7.77%, 11.65%, and 31.07% by dry weight of TPG, respectively. These fractions all showed a single peak analyzed by the Shodex OH-pak SB-804 HQ column (Figure 1D). According to the ^1^H and ^13^C NMR spectra (Figure 2), F1–F3 showed clearer signals than those of the native polysaccharide TPG, and therefore all structural information of TPG could be obtained from analyzing the depolymerized products. Obviously, the smaller the molecular weight of the fractions, the clearer the NMR signals. The ^1^H and ^13^C signals of F2 and F3 were further assigned based on the interpretations of their 2D NMR spectra (Figure 3 and Appendix A).

The anomeric H/C signals at the downfield region in the HSQC spectra of F2 and F3 (Figure 3B,C) were (5.40, 102.36), (5.39, 102.22), (5.37, 102.54), (5.13, 102.87), and (4.97, 100.95 ppm), labeled as A1, B1, C1, A’1, and D1, respectively. Their chemical shifts of H-2 protons can be easily assigned according to the correlation signals to their H-1 protons in the COSY spectra (Appendix A). Other proton chemical shifts, such as H-3, H-4, H-5, and H-6, were assigned by the ^1^H–^1^H COSY, TOCSY, ROESY, and HMBC spectra according to the literature [20,21,22]. Their chemical shifts of carbon could be easily assigned by some correlation spectra, such as ^1^H–^13^C HSQC, HMBC, and TOCSY-HSQC. In the HMBC spectrum, the direct coupling constants (^1^J_C–__H_) of C-1 of each sugar were >170 Hz, indicating that the configurations at C-1 of these sugar residues were of the α-configuration. All the assigned chemical shifts are summarized in Table 2. According to the downfield shift of C-4 (79.42/80.30 ppm), the peak area of the anomeric proton peak of sugar residues, and the proportion in methylation analysis, residue A was confirmed to be →4)-α-D-Glc*p*-(1→. The signal at ~5.39 ppm (residue B) was ascribed to the anomeric proton of the terminal residue α-D-Glc*p*-(1→ from the methylation analysis because of the relatively highfield chemical shifts of its H-4 (3.42 ppm), clearly observed in the TOCSY spectrum (Figure 3A). This terminal residue can be readily identified by the well-distinguished cross-signal (H1, H4) in its TOCSY spectrum. However, this signal has been ignored in many studies on glucans and therefore represent an incorrect assignment [23,24].

Residue C was assigned as (1→4,6)-linked-Glc*p* due to the downfield chemical shifts of C-4 (79.96 ppm) and C-6 (69.95 ppm). The signals of C-4 (~79 ppm) and C-6 (~63 ppm) indicated that both A’ and D were →4)-α-D-Glc*p*-(1→ residue. According to previous studies, the chemical shift of H-1 (4.97 ppm) indicated that the residue D was linked to the C-6 position of the (1→4,6)-linked-Glc*p* residue (residue C) [12,17]. Some H/C signals, such as (3.64/3.69)/65.22 ppm, 3.58/75.31 ppm, 4.01/73.2 ppm, 3.89/84.49 ppm, 3.76/75.32 ppm, and (3.68/3.80)/64.62 ppm were observed and could be assigned to Glc-ol (residue E) at the reducing end. The chemical shift of H-1 of A’ that linked to E appeared at 5.13 ppm. The signals in 1D and 2D NMR spectra of F3 were more distinct than those of F2, indicating that F3 had a higher hydrolytic degree than F2.

The ROESY and HMBC experiments were used to determine the glycosidic linkages of F2 and F3 (Figure 3D, Appendix A). The ROESY spectra clearly showed the inter-residual connectivities between H-1 (5.40 ppm) of residue A and H-4 (3.66/3.67 ppm) of residue A/C, between H-1 (4.97 ppm) of residue D and H-6 (3.86 ppm) of residue C, between H-1 (5.39) of residue B and H-4 (3.66/3.63 ppm) of residue A/D, and between H-1 (5.13) of residue A’ and H-4 (3.89 ppm) of residue E. The correlation signals (3.66, 102.36), (3.89, 102.87), (3.86/3.98, 100.95), and (3.63/3.66, 102.22) in the HMBC spectra indicated that C-1 of residue A linked to C-4 of residue A/C, C-1 of residue A’ linked to C-4 of residue E, C-1 of residue D linked to C-6 of residue C, and C-1 of residue B linked to C-4 of residue A/D. Based on the results of methylation and NMR analysis, the main chain of F2 and F3 was →4)-α-D-Glc*p*-(1→ repeating units, and the branches were composed of disaccharide units of α-D-Glc*p*-(1→4)-α-D-Glc*p*-(1→ linked to approximately every four Glc*p* moieties through their C-6 position. The ratio of sum of peak areas of →4)-Glc*p*-(1→ and →4,6)-Glc*p*-(→ to that of Glc*p*-(1→ was about 5:1 (Table 1), and approximate integration of the anomeric signals in the ^1^H NMR spectrum of TPG gave the molar ratio of residues (A+B+C) and D of about 5:1, indicating that disaccharide branches occurred every four Glc*p* units along the backbone. A possible structure for F2 and F3 was proposed and is shown in Figure 3E. TPG had the same structural feature as that of F2 and F3. 

To date, some α-glucans with different structures from various sources have been obtained. Many mushrooms, animals, and plants, such as *Dictyophora echinovolvata*, *Ganoderma capense*, *Sinonovacula constricta*, *Epimedium koreanum* Nakai, and *Isatis indigotica* contain glucans having a linear α-(1→4)-D-Glc*p* backbone and a α-D-Glc*p*-(1→ branch directly linked to the backbone at C-6 with different degrees of branch [12,22,23,24,25]. A novel glucan was also isolated from *Crataegus pinnatifida*, which was composed of a linear α-(1→4)-D-Glc*p* backbone and a 1,6-linked α-Glc*p* disaccharide side chain linked to C-6 of the backbone [26]. Glucans also widely exist in marine animals. For example, a (1→4)-α-D-Glc*p* backbone, with branching positions located at *O*-3 of the backbone with a terminal-α-D-Glc*p,* was isolated from oysters (*Crassostrea gigas*) [27]. These glucans with various structures showed good activities, such as immunostimulatory, antioxidant, and anti-tumor activities. In our present study, the TPG isolated from *T. palmiferus* was a novel glucan. As far as we know, no reported glucans isolated from natural products possess the same structure as that of TPG.

### 2.3. Physicochemical Properties of TPGS

TPGS was prepared with a yield of 80% based on the dry weight of TPG. The profile of TPGS analyzed by the Shodex OHpak SB-804 HQ column is shown in Figure 1A, and the *M*_w_ was calculated as 142.2 kDa. The retention time of TPGS was reduced compared with TPG, and two small peaks near the peak of H_2_O were observed, indicating that TPG was degraded in the process of sulfation. The contents of total carbohydrate and sulfate groups were determined to be 49.7% and 46.7%, respectively. The total carbohydrate content of TPGS was significantly lower than that of TPG, which was due to the introduction of sulfate groups [28]. The degree of sulfation (DS) was determined as 1.57.

The FT-IR spectrum of TPGS is presented in Figure 1C, where it can be observed that two new characteristic absorption bands appear in the spectrum of the sulfated derivative TPGS. The strong absorption peak at about 1231.61 cm^−1^ was attributed to S=O stretching vibration. The peak appeared at around 816.50 cm^−1^ was the C-O-S stretching vibration [29]. 

The sulfate group results in the downfield chemical shifts of proton and carbon at the substituent site. Therefore, the positions of sulfate ester substituents on each sugar residue could be confirmed in comparison with corresponding sugar residue in the TPG. According to ^1^H–^13^C HSQC spectra and chemical shifts of TPG and TPGS (Figure 4 and Appendix A), all the hydroxyl groups at C-2 and C-3 of residues B, C, and D were sulfated. In the backbone, all the hydroxyl groups at C-2 of residue A were sulfated, but only partial hydroxyl groups at its C-3 positions were sulfated due to some signals appearing at (4.07, 76.19 ppm) from A3 in TPGS near to those (3.97, 75.98 ppm) in TPG in the HSQC spectrum (Figure 4). All the hydroxyl groups at C-4 of residue B were sulfated, and most of the C-6 positions of residues A, B, and D were sulfated. Based on these results, the OH-2, OH-4, and CH_2_OH-6 groups were preferentially sulfated, followed by that of OH-3, which was consistent with the results in previous studies [30]. The 2D NMR spectra of TPGS showed weak or overlapping signals hampering resolution, as expected for high-molecular-weight sulfated polysaccharides. The detailed structures and sulfated patterns of TPGS still need to be elucidated by preparing and analyzing its depolymerized products in the future.

In previous studies, sulfated modification of polysaccharides from various sources, such as Grifola frondose, Cladonia ibitipocae, and Cyclocarya paliurus, was performed to prepare sulfated polysaccharides with high bioactivities [30,31,32]. However, these polysaccharides are heteropolysaccharides consisting of several monosaccharides, such as arabinose, galactose, glucose, and mannose, and have very complex structures. Therefore, the structures of their sulfates are not as clear as that of TPGS, which may limit their further research and application as new drugs. In this study, TPG and TPGS had a relatively regular structure, which may have high potential to be lead compounds of drugs.

### 2.4. Analysis of Anticoagulant Activity of TPGS

Anticoagulant activities were measured by APTT, PT, and TT, which are commonly used to evaluate the ability of a compound to inhibit blood clotting [33]. The concentration of TPG required for double APTT, PT, and TT was higher than 128 µg/mL (not shown), indicating that TPG had no anticoagulant activity. The concentration of TPGS needed to double APTT was 4.26 μg/mL, demonstrating potent activity in inhibiting the intrinsic coagulation pathway (Figure 5). Notably, this activity was close to that of the low-molecular-weight heparin (LMWH), enoxaparin (3.54 μg/mL), but lower than that of unfractionated heparin (HP) (0.82 μg/mL). TPGS showed weak PT-prolonging activity because of the required concentration for doubling PT (30.88 μg/mL), significantly higher than that of HP (3.35 μg/mL), by about 10 times. The concentration required to double TT was 14.12 μg/mL, indicating that TPGS had a potent effect on the common coagulation pathway. 

The EC_50_ value of TPGS for inhibiting intrinsic tenase (FXase) was 77.15 ng/mL, which was close to that of LMWH (69.82 ng/mL). LMWH showed strong AT-dependent anti-FXa and anti-FIIa activities with EC_50_ values of 61.49 and 46.07 ng/mL, respectively, which was consistent with a previous study [34]. However, TPGS showed no AT-dependent anti-FIIa and anti-FXa activities (EC_50_ > 2000 ng/mL). Based on our present study, TPGS exerted anticoagulant activity by inhibiting the intrinsic FXase, and the effects of TPGS on the extrinsic and common pathways should be further investigated.

The existing clinically used anticoagulant agents have the side effect of increasing bleeding risk. Inhibitors that act on the intrinsic clotting pathway may have low bleeding tendency and have become a research hotspot in searching for new anticoagulant drugs. Intrinsic FXase is a significant anticoagulant target, and inhibitors selectively inhibiting this enzyme complex have low bleeding tendency [8]. Our previous studies on oligosaccharides from sea cucumber supported that strongly inhibiting the FXase did not show serious bleeding risk [7]. The molecular weight, content, and substitution position of the sulfate group and carboxyl group in the oligosaccharides affected the anticoagulant activity [35]. In this study, TPG did not have anticoagulant activity, but TPGS could effectively inhibit the FXase. These results indicate that the sulfate group had a decisive influence on the anticoagulant activity, which might be related to the high level of negative charge produced by sulfate groups. TPGS had higher selectivity to FXase than LMWH, since LMWH could inhibit FXa and FIIa. This may be due to the sulfated disaccharide side chains contained in the TPGS, which is consistent with that of the oligosaccharides from sea cucumber [33]. Therefore, TPGS may have lower bleeding risk. Given that HP and LMWH extracted from cattle, pigs, and other mammals have the risk of contamination by viruses and pathogens during the preparation process, TPGS may be a potentially safe lead compound of anticoagulant drugs. The specific structure–activity relationships of TPGS need to be further investigated in the future. 

## 3. Materials and Methods

### 3.1. Materials

Brittle stars were obtained from a local market of Beihai City in Guangxi Zhuang Autonomous Region of China and identified to be *T. palmiferus* by Dr. Shengping Zhong. TFA, bovine serum albumin (BSA), L-fucose (Fuc), and D-mannose (Man) were obtained from Aladdin Chemical Reagent Co., Ltd. (Shanghai, China). N-acetyl-D-glucosamine (GlcNAc), D-glucuronic acid (GlcA), D-Glc*p*, N-acetyl-D-galactosamine (GalNAc), D-galacturonic acid (GalA), D-galactose (Gal), and 1-phenyl-3-methyl-5-pyrazolone (PMP) were purchased from Sigma-Aldrich (St. Louis, MO, USA). Papain (800 U/mg) was purchased from Shanghai Yuanye Bio-Technology Co., Ltd. (Shanghai, China). Amberlite FPA98 Cl ion-exchange resin, Sepharose CL-6B, and Bio-gel P6 were from Rohm and Haas Company (Philadelphia, PA, USA), GE Healthcare Life Sciences (Uppsala, Sweden), and Bio-Rad Laboratories (Hercules, CA, USA), respectively. APTT, TT, and PT reagents, CaCl_2_ solution (0.05 M), and coagulation control were obtained from TICO GmbH (Hamburg, Germany). Enoxaparin (0.4 mL × 4000 AXaIU) and human FVIII were from Sanofi-Aventis (Paris, France) and Bayer Healthcare LLC (Leverkusen, Germany), respectively. Thrombin substrate (CS01(38)), Biophen FVIII: C kit, heparin anti-FIIa kit, and heparin anti-FXa kit were purchased from Hyphen Biomed (France). All other chemicals, such as H_2_SO_4_, CHCl_3_, and HCl, were of analytical grade and are commercially available on the market.

### 3.2. Extraction, Isolation, and Purification of TPG

The extraction of polysaccharides from *T. palmiferus* was based on our previous method, with some modifications [34]. The brittle stars were dried in an oven at 60 °C and then pulverized into a uniform powder. The obtained powder was enzymatically digested with 0.1% papain aqueous solution at 55 °C for 16 h. The reaction mixture was then extracted with 0.5 M NaOH at 60 °C for 2 h and centrifuged at 4816× *g* for 15 min. After addition of 6 M HCl until pH 3 was reached, the solution was kept at 4 °C for 4 h and centrifuged at 4816× *g* for 15 min to remove proteins. The obtained supernatant was neutralized with 6 M NaOH solution. The ethanol was added in the supernatant to a final concentration of 75% (*v*/*v*), and the precipitate obtained after centrifugation was dissolved in deionized water. The polysaccharide solution was decolorized by using 3% H_2_O_2_ aqueous solution (pH 10) at 50 °C. After decolorization, the polysaccharides were precipitated with ethanol as stated above. After centrifugation and lyophilization, the crude polysaccharides were obtained. 

The crude polysaccharides were purified by ion-exchange chromatography with a Amberlite FPA98 column and gel filtration with a Sepharose CL-6B column (1.5 cm × 150 cm). The collected fractions were detected by phenol-sulfuric acid method. The neutral polysaccharide was obtained, dialyzed with a dialysis bag (cut-off 3500 Da), and lyophilized.

### 3.3. Preparation of Depolymerized Product of TPG

TPG was depolymerized according to a previous method with minor modifications [36]. Briefly, TPG (103 mg) was dissolved in 10.3 mL of 100 mM trifluoroacetic acid and hydrolyzed at 100 °C for 7 h in an oil bath. The reaction mixture was then adjusted to pH 7.0 with 1 M NaOH, added NaBH_4_ (39 mg), and incubated at 50 °C for 1 h. After the reaction, the solution was neutralized by HCl, concentrated, and freeze-dried to obtain the depolymerized product dTPG. The dTPG was then separated by gel filtration with a Bio-Gel P-6 Gel column.

### 3.4. Sulfated Modification of TPG

The TPGS was prepared by using SO_3_·pyridine (SO_3_·Py) complex reagent [37]. TPG (100 mg) was stirred in anhydrous DMSO until dissolution. The SO_3_·Py complex was added and stirred at 50 °C for 1 h. After the reaction, the mixture was precipitated with anhydrous alcohol. Subsequently, the polysaccharide sulfate TPGS was purified by ion-exchange chromatography with a DEAE Sepharose Fast Flow column. The fractions eluted by 2.0 M NaCl solution were collected, concentrated, desalted with a dialysis bag (cut-off 3500 Da), and finally lyophilized. DS was calculated according to the following formula: DS = [(1.62 × S%)/(32 − 1.02 × S%)]
where S% is the mass fraction of sulfur atom.

### 3.5. Physicochemical Properties 

The content of total carbohydrate in polysaccharides was analyzed by the phenol-sulfuric acid method established by Dubois et al., using Glc as the standard [38]. The protein content was determined by the Coomassie brilliant blue method as described by Bradford et al., using BSA as the standard [39]. A classical turbidimetric method was used to measure the content of sulfuric radical [40]. 

The molecular weight was determined by using a Shimadzu LC-2030C 3D HPLC apparatus (Shimadzu Corp., Kyoto, Japan) with a refractive index RID detector and analyzed by a Shodex OHpak SB-804 HQ column (7 µm, 8 × 300 mm) [34]. The column temperature was maintained at 35 °C. The mobile phase was 0.1 M NaCl solution at a flow rate of 0.5 mL/min. The molecular weight estimation was calibrated by pullulan standards with molecular weights of 344.0, 107.0, 47.1, 21.1, and 9.6 kDa. 

Analysis of monosaccharide compositions was performed according to our previous study on the LC-2030C 3D HPLC apparatus with a DAD detector and an Eclipse Plus C18 column (5 μm, 4.6 × 250 mm) [41]. Briefly, samples were hydrolyzed by using 4.0 M TFA at 120 °C for 2 h. The hydrolysate was labeled with PMP under alkaline condition and analyzed by the HPLC at 245 nm. The mobile phase was constituted by 83% phosphate-buffered saline (0.1 M, pH 6.7) and 17% acetonitrile (*v*/*v*) at a flow rate of 1.0 mL/min. The column temperature was maintained at 30 °C.

The UV-vis absorption spectra were recorded on an Evolution 350 spectrophotometer (Thermo) in the wavelength range of 190–800 nm. The dried polysaccharide samples (1–2 mg) were mixed with dried KBr pellets and pressed into transparent sheets. The FT-IR spectra were determined on a Nicolet iS50 Fourier-transform infrared spectroscopy spectrometer (Thermo Fisher Scientific, Waltham, MA, USA) in the range of 400–4000 cm^−1^.

The determination of NMR spectra was performed on a Bruker Advance 600 MHz spectrometer with the ^1^H/^13^C dual probe in FT mode at 298.1 K [42]. The dried samples were dissolved in 0.5 mL of deuterium oxide (D_2_O, 99.9% D) and lyophilized three times to replace exchangeable protons with D_2_O. The dried polysaccharide samples were then dissolved in 0.5 mL of D_2_O at a concentration of 10–20 g/L for NMR analysis.

### 3.6. Methylation and GC-MS Analysis

The glycosidic linkage patterns of TPG were analyzed by methylation using the method described previously [25]. In detail, the dried polysaccharide (5 mg) was added into anhydrous DMSO (2 mL). When the sample was dissolved, the anhydrous NaOH (approximately 50 mg) was added under nitrogen atmosphere. After a few minutes in an ice bath, the iodomethane (1.5 mL) was slowly added and stirred for 2 h in darkness. The reaction was then terminated by deionized water (2 mL) and extracted by an equal volume of chloroform. These procedures were repeated several times to obtain the methylated polysaccharide and analyzed by FT-IR. After exhaustive methylation, the polysaccharide was hydrolyzed with 2 M TFA at 120 °C for 2 h, reduced by NaBH_4_ (30 mg) for 3 h at room temperature, and reacted with acetic anhydride and pyridine at 100 °C for 1 h. The acetylated derivatives were extracted with dichloromethane. The organic phase was evaporated, dissolved in dichloromethane, and then analyzed by GC-MS. The Shimadzu GCMS-QP 2010 was equipped with a RXI-5 SIL MS column (30 m × 0.25 mm × 0.25 μm). The heating conditions of the program were as follows: the initial temperature was 120 °C, the temperature was raised 3 °C/min to 250 °C and maintained for 5 min. Both the inlet temperature and the detector temperature were 250 °C. The carrier gas was helium at the flow rate of 1 mL/min.

### 3.7. Assays of Anticoagulant Activities

In accordance with previous studies, APTT, TT, and PT of TPG and TPGS were determined by using a coagulometer (TECO MC-2000, Hamburg, Germany) according to the manufacturer’s instructions [7]. Briefly, 1.28 mg of the sample was accurately weighed, dissolved in Tris-HCl buffer solution (pH 7.4) to prepare 1280 μg/mL solution, and diluted with Tris-HCl buffer solution (pH 7.4) to obtain a series of concentrations of samples.

For the APTT assay, the sample (5 μL) was added to the test tube prewarmed at 37 °C. Standard human plasma (45 μL) was added and incubated at 37 °C for 2 min. APTT reagent (50 μL) preheated at 37 °C was added and incubated at 37 °C for 3 min. Finally, 0.02 M CaCl_2_ (50 μL) prewarmed at 37 °C was added, and the timing was initiated to record the clotting time.

For the TT determination, different concentrations of polysaccharides (10 μL) diluted with 20 mM Tris-HCl buffer (pH 7.4) were mixed with standard human plasma (90 μL) in a test tube prewarmed at 37 °C and incubated for 2 min. The mixture was then added TT reagent (50 μL) preheated at 37 °C, and the clotting time was recorded at the same time. 

For the PT assay, sample solutions at various concentrations (5 μL) were added to the detection tube, and standard human plasma (45 μL) was added and incubated at 37 °C for 2 min. Finally, PT reagent preheated at 37 °C was added, and the coagulation time started to record immediately.

### 3.8. Measurement of Inhibition of Human Intrinsic Factor Xase

The inhibition of intrinsic FXase was measured by using the BIOPHEN FVIII: C kit according to our previous study [34]. Briefly, 30 μL of the polysaccharide solution at a gradient concentration, 2 IU/mL factor VIII (30 μL), and 60 nM factor IXa (30 μL) (containing FIIa, Ca^2+^, and PC/PS) were added to the 96-well plate and incubated for 2 min at 37 °C. The reaction was activated by 50 nM factor X (30 μL) containing FIIa inhibitor, and the plate was incubated at 37 °C for 1 min. Finally, after addition of 8.40 mM chromogenic substrate SXa-11 (30 μL), the amount of factor Xa generation was calculated by the absorbance at 405 nm every 30 s continuously for 5 min at 37 °C. 

### 3.9. Determination of Inhibition of Human FIIa and FXa

In the presence of AT, the anti-FIIa and anti-FXa activities of samples were measured by the BIOPHEN Heparin Anti-FIIa kits and Anti-FXa kits, respectively, according to the method of our previous study [7]. In detail, Tris-HCl (pH 7.4) buffer was used as a control. Different concentrations of samples (30 μL) were added 30 μL of R1 solution (1 IU/mL AT solution for anti-FXa assay, and 0.25 IU/mL AT solution for anti-FIIa detection, respectively) into the 96-well microplates, mixed, and incubated for 1–2 min at 37 °C (1 min for anti-FXa detection and 2 min for anti-FIIa measurement). Then, 30 μL of R2 solution (24 IU/mL FIIa for anti-FIIa detection and 8 μg/mL FXa solution for anti-FXa detection) was added, mixed, and incubated accurately at 37 °C for 1–2 min (1 min for anti-FXa determination and 2 min for anti-FIIa determination). Prewarmed R3 solution (1.2 mmol/L FXa specific chromogenic substrate CS-11 (65)) for anti-FXa measurement and 1.25 mmol/L factor IIa specific chromogenic substrate CS-01 (38) for anti-FIIa determination) (30 μL) was add to the reaction mixture. After mixing, the absorbance at 405 nm was read every 30 s for 5 min at 37 °C, and the enzyme activity was expressed by the change rate of absorbance.

## 4. Conclusions

In this study, the crude polysaccharides of brittle stars were extracted by alkaline hydrolysis combined with enzymatic hydrolysis after grinding, and a high purity polysaccharide component TPG was obtained after separation and purification. The dTPG was prepared by trifluoroacetic acid hydrolysis, the fractions with different *M*_w_ were separated, and their structures were elucidated by NMR spectroscopy data interpretation. The structure of TPG was deduced by the fractions to be a novel glucan, which consisted of a linear α-(1→4)-Glc*p* backbone and a disaccharide side chain of α-D-Glc*p*-(1→4)-α-D-Glc*p*-(1→ linked to every four Glc*p* moieties at the C-6 position in the backbone. TPG was then modified by a sulfur trioxide-pyridine complex method, producing a sulfated derivative named as TPGS. The OH-2, OH-4, and OH-6 groups were preferentially sulfated, followed by that of OH-3. After the sulfated modification, the anticoagulant activity significantly increased, indicating that sulfate group was beneficial to enhance the anticoagulant activity of TPGS. TPGS showed strong anti-FXase activity, with an EC_50_ value of 77.15 ng/mL, and had higher selectivity to inhibit FXase than LMWH, which may show low bleeding tendency. The different action mechanism between TPGS and LMWH may contribute to their different structural features, such as monosaccharide composition and sulfated disaccharide side chains. Hence, TPGS is expected to be used as a potential safe lead compound of anticoagulant drugs and will be useful for investigating the structure–activity relationship of sulfated polysaccharides. In the future, more research is required to elucidate the detailed mechanisms of the anticoagulant activities of TPGS.

## Figures and Tables

**Figure 1 marinedrugs-21-00148-f001:**
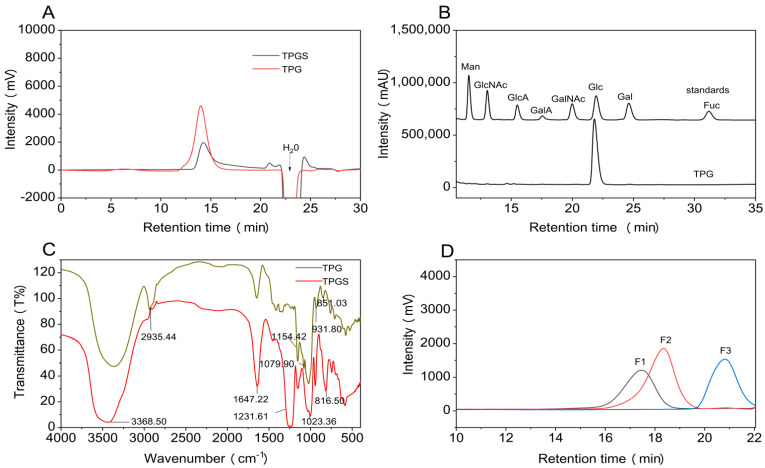
Physicochemical properties of TPG, TPGS, and fractions F1–F3. HPGPC profiles of TPG, TPGS (**A**), and fractions F1–F3 (**D**), chromatograms of PMP derivatives of mixed monosaccharide standards and TPG (**B**), and FT-IR spectra of TPG and TPGS (**C**).

**Figure 2 marinedrugs-21-00148-f002:**
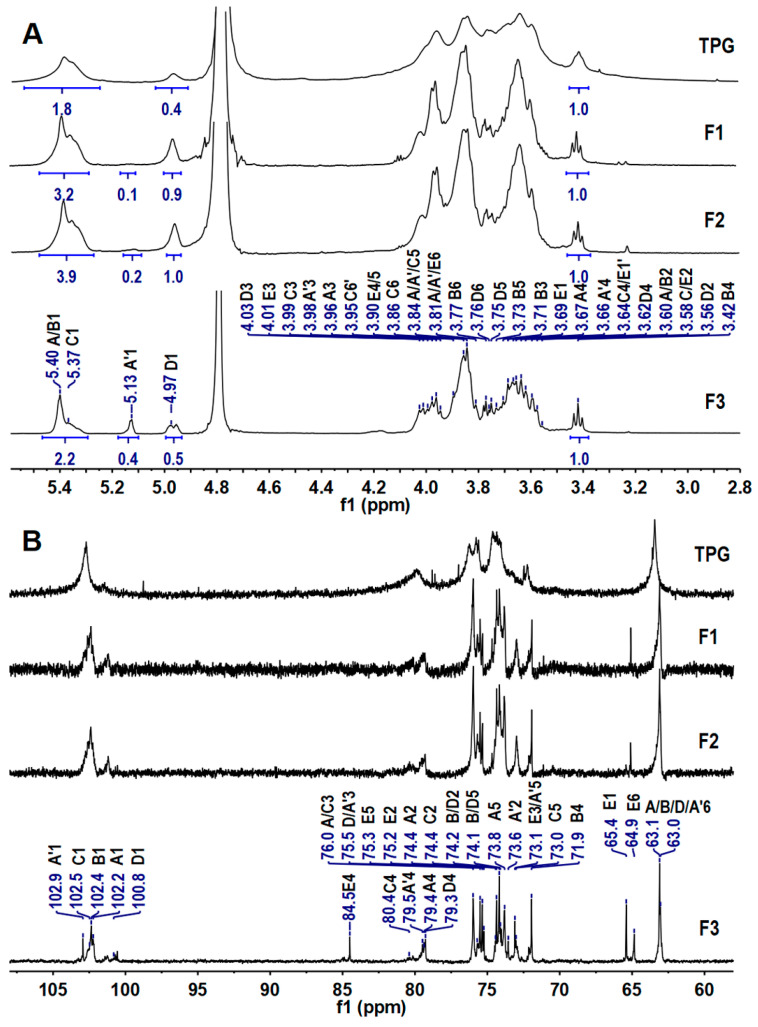
^1^H (**A**) and ^13^C (**B**) NMR spectra of TPG and fractions F1–F3.

**Figure 3 marinedrugs-21-00148-f003:**
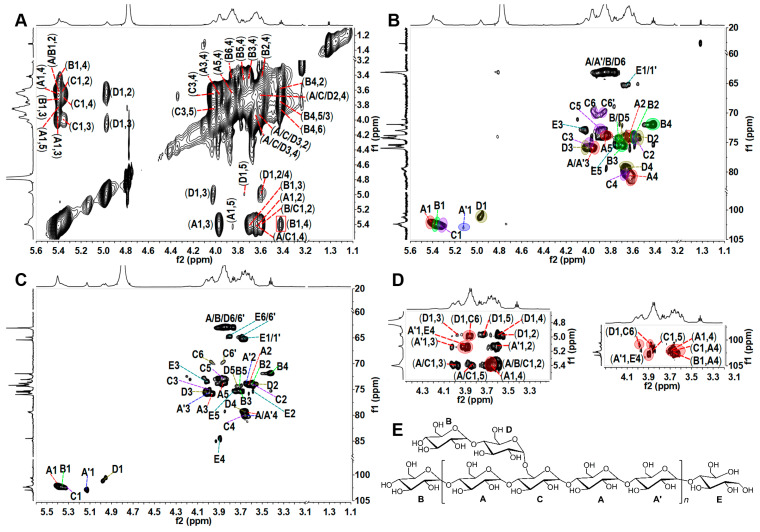
2D NMR spectra of F2 and F3 (**A**–**D**), and a proposed structure of F2 and F3 (**E**). (**A**,**B**) are TOCSY and HSQC spectra of F2, (**C**) is HSQC spectrum of F3, and (**D**) is the partial ROESY and HMBC spectra of F3. The signals labeled in red ellipses in (**D**) indicate the connection positions.

**Figure 4 marinedrugs-21-00148-f004:**
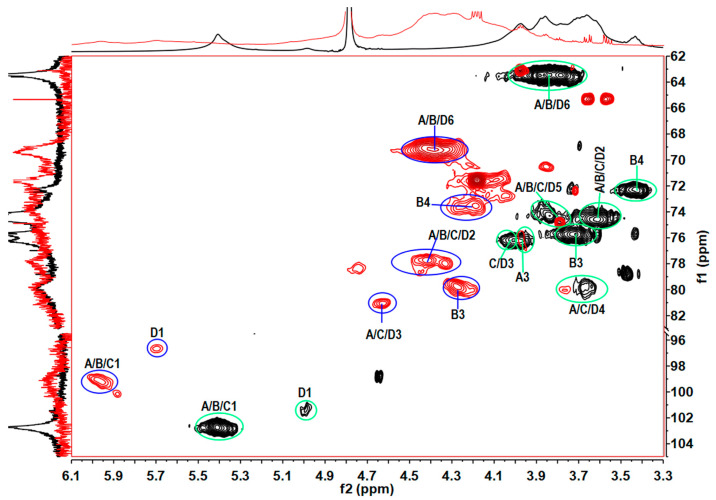
Superimposed ^1^H–^13^C HSQC spectra of TPG (black) and TPGS (red).

**Figure 5 marinedrugs-21-00148-f005:**
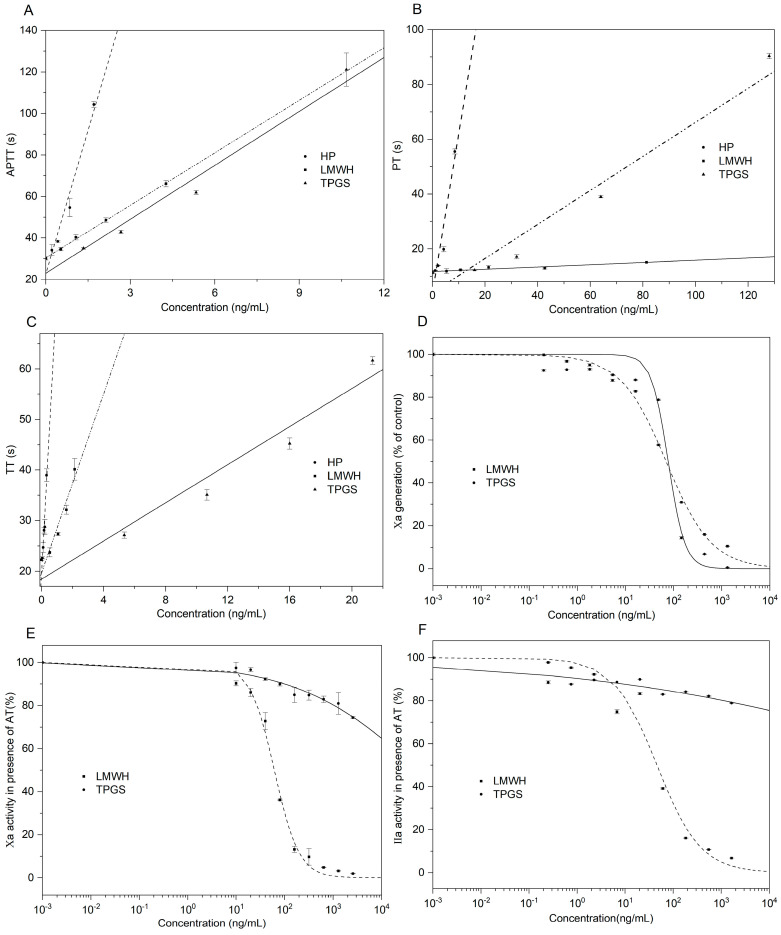
Effects of TPGS on APTT (**A**), PT (**B**), TT (**C**), intrinsic FXase (**D**), FXa (**E**), and FIIa (**F**) (n = 3).

**Table 1 marinedrugs-21-00148-t001:** Results of methylation analysis of TPG.

Relative Retention Time (min) ^a^	Methylated Sugars	Mass Fragments (*m*/*z*)	Type of Linkage	Area (%)
1.00	2,3,4,6-Me_4_-Glc*p*	43, 71, 87, 101, 117, 129, 145, 161, 205	Glc*p*-(1→	17.2
1.25	2,3,6-Me_3_-Glc*p*	43, 87, 99, 101, 113, 117, 129, 131, 161, 173, 233	→4)-Glc*p*-(1→	69.66
1.40	2,6-Me_2_-Glc*p*	43, 87, 97, 117, 159, 185	→3,4)-Glc*p*-(→	0.61
1.53	2,3-Me_2_-Glc*p*	43, 71, 85, 87, 99, 101, 117, 127, 159, 161, 201	→4,6)-Glc*p*-(→	12.53

^a^ Relative retention time of 2,3,4,6-Me_4_-Glc*p* was set to 1.

**Table 2 marinedrugs-21-00148-t002:** ^1^H and ^13^C NMR chemical shifts of F2.

Residues	H/C	Chemical Shifts (δ, ppm)					
		1	2	3	4	5	6
**A**	H	5.40	3.61	3.97	**3.66 ^a^**	3.85	3.81/3.91
	C	102.36	74.41	75.98	**79.42/80.30**	73.90	63.11
**B**	H	5.39	3.60	3.71	3.42	3.73	3.77/3.85
	C	102.22	74.07	75.47	71.89	74.15	63.11
**C**	H	5.37	3.58	4.00	**3.67**	3.87	**3.86/3.98**
	C	102.54	74.35	75.94	**79.96**	73.01	**69.95**
**A’**	H	5.13	3.61	3.98	**3.63**	3.85	3.81/3.91
	C	102.87	73.60	75.82	**79.85**	73.10	63.11
**D**	H	4.97	3.59	4.03	**3.63**	3.75	3.76/3.84
	C	100.95	74.00	75.85	**79.35**	74.14	63.19
**E**	H	3.64/3.69	3.58	4.01	**3.89**	3.76	3.68/3.80
	C	65.22	75.31	73.2	**84.49**	75.32	64.62

^a^ Values in boldface indicate glycosylated positions.

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
