# Peer review of "Structural Elucidation of a Glucan from Trichaster palmiferus by Its Degraded Products and Preparation of Its Sulfated Derivative as an Anticoagulant"

_marinedrugs, 2023, doi:10.3390/md21030148_

Round 1
Reviewer 1 Report
In the manuscript isolation, structural characterization, chemical modifications of glucan (TPG) from the brittle star Trichaster palmiferus is described. The structure of the parent polysaccharide and its derivatives was elucidated by a number of physicochemical methods including NMR. It was shown that TBG composed of α1,4–linked D-glucose (Glc) backbone together with α1,4–linked Glc disaccharide side chain linked through C-1 to C-6 of the main chain. The TPG sulfate (TPGS) was prepared, and its anticoagulant activity was studied in vitro. Results of anticoagulant activity showed that TPGS significantly prolonged activated partial thromboplastin time. Furthermore, TPGS obviously inhibited intrinsic tenase with EC50 value of 77.15 ng/mL, which was comparable to that of low-molecular-weight heparin (LMWH) (69.82 ng/mL). TPGS showed no AT-dependent anti-FIIa and anti-FXa activities.
Notes
1. The Table summarized the NMR data of TPGS is required.
2. The values of SD should be indicated on all plots in Fig. 5.
3. The yields of TPG and its derivatives should be pointed in Experimental section.
4. Section 3.7. The concentration of the tested sample and the volume of the plasma should be indicated.
5. Lines 235-237. I don’t agree with the statement “TPGS showed strong PT-prolonging activity, and the required concentration for doubling PT (30.88 μg/mL) was lower than that of HP (3.35 μg/mL) about 10 times”. 30.88 μg/mL is about 10 times higher than 3.35 μg/mL. Please, analyze this result again.
Author Response
Thanks very much for comments concerning our manuscript. These comments are valuable and helpful for revising and improving our paper. We have studied the comments carefully and have made changes to the manuscript in RED. The responses to the comments are as follows:
1. The Table summarized the NMR data of TPGS is required.
Response: Thanks very much for your suggestion. The NMR data of TPGS have been summarized and supplemented in Supplementary Materials of the revised manuscript.
2. The values of SD should be indicated on all plots in Fig. 5.
Response: Thanks for your suggestion. In Figure 5, the size of the points is large and the error bar is small resulting in that the error bar cannot observed in the plots. We have narrowed the points down in Figure 5.
3. The yields of TPG and its derivatives should be pointed in Experimental section.
Response: Thanks for your suggestion. We have provided this information in the new manuscript.
4. Section 3.7. The concentration of the tested sample and the volume of the plasma should be indicated.
Response: Thanks very much for your suggestion. We have added the concentration of the tested sample and the volume of the plasma in Section 3.7. of the new manuscript.
5. Lines 235-237. I don’t agree with the statement “TPGS showed strong PT-prolonging activity, and the required concentration for doubling PT (30.88 μg/mL) was lower than that of HP (3.35 μg/mL) about 10 times”. 30.88 μg/mL is about 10 times higher than 3.35 μg/mL. Please, analyze this result again.
Response: Thanks very much for your helpful suggestion. We agree with your suggestion. The PT-prolonging activity of TPGS is week due to the required concentration of 30.88 μg/mL for doubling PT. We have reanalyzed this result carefully in the new manuscript.
Reviewer 2 Report
The title manuscript reports the extraction, purification and structural characterization of a branched a-1®4-glucan from a brittle star, as well as a preliminary in vitro evaluation of the anti-coagulant activity of a sulfated derivative thereof. The topic is interesting and fits very well the aims and scope of Marine Drugs. The manuscript is well written, data are clearly presented and discussed. My only concern regards some conclusions that in my opinion are not supported enough by the reported results. In particular:
i) the authors state that disaccharide branches occur every five Glc units along the polysaccharide backbone (Figure 3E, L170-173 and L425-427). How was the number five derived? PMAA data reported in Table 1 (area of peaks related to 4-linked-Glc and 4,6-linked-Glc units vs. t-Glc one) as well as 1H-NMR integration values in Figure 2A (ratio between signal at 5.4 and 5.0 ppm in native polysaccharide spectrum) display a ratio between 4- + 4,6-linked-Glc and t-Glc units equal to five. Taking into consideration that one 4-linked-Glc residue is located in the branch, this suggests that disaccharide branches occur every four rather than five Glc units along the backbone;
ii) the authors state a preferential sulfation at O-2, O-4 and O-6 sites over the O-3 positions (L216-218 and L429-L430). In my opinion this cannot be stated exclusively by comparison of the 1H,13C-HSQC spectra of the native and sulfated polysaccharide. A more extensive structural characterization of the latter with the acquisition and analysis of more 2D-NMR spectra is mandatory.
Some minor points also needing a slight revision:
- table 1, row 2, column 3: modify “4371” into “43, 71”;
- figure 3: why the NMR spectra display also a small slice of the region at d 1.2-1.4/20-30 ppm, although no significant signals are present there?;
- L153-155: a reference should be added to support the statement;
- L365: modify “iodide methane” into “iodomethane”;
- L432: is the term “antiglycation” correct?
In conclusion, the manuscript can be accepted for publication in Marine Drugs after the authors have revised the points indicated above.
Author Response
Thanks very much for comments concerning our manuscript. These comments are valuable and helpful for revising and improving our paper. We have studied the comments carefully and have made changes to the manuscript in RED. The responses to the comments are as follows:
i) the authors state that disaccharide branches occur every five Glc units along the polysaccharide backbone (Figure 3E, L170-173 and L425-427). How was the number five derived? PMAA data reported in Table 1 (area of peaks related to 4-linked-Glc and 4,6-linked-Glc units t-Glc one) as well as 1H-NMR integration values in Figure 2A (ratio between signal at 5.4 and 5.0 ppm in native polysaccharide spectrum) display a ratio between 4- + 4,6-linked-Glc and t-Glc units equal to five. Taking into consideration that one 4-linked-Glc residue is located in the branch, this suggests that disaccharide branches occur every four rather than five Glc units along the backbone;
Response: Thanks very much for your valuable comments on our manuscript. We have revised the proposed structure in Figure 3 and analyzed these results in detail in the main text according to your good suggestions.
ii) the authors state a preferential sulfation at O-2, O-4 and O-6 sites over the O-3 positions (L216-218 and L429-L430). In my opinion this cannot be stated exclusively by comparison of the 1H,13C-HSQC spectra of the native and sulfated polysaccharide. A more extensive structural characterization of the latter with the acquisition and analysis of more 2D-NMR spectra is mandatory.
Response: Thanks for your comments. The sulfate group results in the down-field chemical shifts of proton and carbon at the substituent site. Therefore, the positions of sulfate ester substituents on each sugar residue could be confirmed in comparison with corresponding sugar residue in the TPG. The 1H-1H TOCSY spectrum and 1H and 13C NMR chemical shifts of TPGS are provided in Figure S12 and Table S1, respectively. The chemical shifts of proton and carbon of OH-2, OH-4, and CH2OH-6 shift to down-field obviously. However, some signals appear at (4.07, 76.19 ppm) from A3 in TPGS near to those (3.97, 75.98 ppm) in TPG in the HSQC spectrum (Figure 4). Therefore, the OH-2, OH-4, and CH2OH-6 groups were preferentially sulfated, followed by that of OH-3. The 2D NMR spectra of TPGS showed weak or overlapping signals hampering resolution, as expected for high-molecular-weight sulfated polysaccharides. The detailed structures and sulfated patterns of TPGS still need to be elucidated by preparing and analyzing its depolymerized products in the future. We have added these discussions in the revised manuscript.
Some minor points also needing a slight revision:
- table 1, row 2, column 3: modify “4371” into “43, 71”;
Response: We have changed “4371” to “43, 71” in the revised manuscript.
- figure 3: why the NMR spectra display also a small slice of the region at d 1.2-1.4/20-30 ppm, although no significant signals are present there?;
Response: Thanks for your question. Although the signals appeared at 1.2-1.4 ppm, and 20-30 ppm were too weak to be assigned, we show all the signals appeared in the NMR spectra as far as we can. It can confirm the high purity of our polysaccharides.
- L153-155: a reference should be added to support the statement;
Response: Thanks for your suggestion. We have added two references to support the statement.
- L365: modify “iodide methane” into “iodomethane”;
Response: We have modified “iodide methane” into “iodomethane” in the new manuscript.
- L432: is the term “antiglycation” correct?
Response: We are ashamed of this mistake. The “antiglycation” has been changed to “anticoagulant” in the new manuscript.
Reviewer 3 Report
This research article (marinedrugs-2217221) focused on the structural elucidation of a glucan from a novel source (T. palmiferus) by its degraded products and preparation of its sulfated derivative as an anticoagulant. The structure of degraded products of the glucan (TPG) was determined by physicochemical analysis and detailed NMR characterization. TPG was then deduced to be a new glucan with disaccharide side chains. The TPG sulfate (TPGS) prepared based on the regular TPG showed strong anticoagulant activity and potent anti-FXase activity with EC50 value of 77.15 ng/mL and had higher selectivity to inhibit FXase than LMWH. The sulfated disaccharide side chains may affect anticoagulant activities of the TPGS. The research had novelty, which laid the foundation for exploring the structure-activity relationship of the anticoagulant activity of sulfated polysaccharides. The article is clearly written, the characterization complete and well performed, and the anticoagulant activity of the TPGS interesting. I believe that the article can be published after some details that I explain below been improved.
1. In Line 34, the “molluscs” should be changed to “mollusks”.
2. In Line 38 and 221, a comma before “and” could make the sentence clearer.
3. In Line 48, the “Antartic” should be changed to “Antarctic”.
4. In Line 329, the “coomassie” should be changed to “Coomassie”.
5. In Line 346, the “EvoLution” should be changed to “Evolution”.
Author Response
Thanks very much for comments concerning our manuscript. These comments are valuable and helpful for revising and improving our paper. We have studied the comments carefully and have made changes to the manuscript in RED. The responses to the comments are as follows:
1. In Line 34, the “molluscs” should be changed to “mollusks”.
Response: Thanks for your suggestions. We have changed the “molluscs” to “mollusks”.
2. In Line 38 and 221, a comma before “and” could make the sentence clearer.
Response: We have added a comma before “and” in Line 38 and 221.
3. In Line 48, the “Antartic” should be changed to “Antarctic”.
Response: We have have changed the “Antartic” to “Antarctic”.
4. In Line 329, the “coomassie” should be changed to “Coomassie”.
Response: We have have changed the “coomassie” to “Coomassie”.
5. In Line 346, the “EvoLution” should be changed to “Evolution”.
Response: We have have changed the “EvoLution” to “Evolution”.